# The Potential Harmful Effects of Genetically Engineered Microorganisms (GEMs) on the Intestinal Microbiome and Public Health

**DOI:** 10.3390/microorganisms12020238

**Published:** 2024-01-23

**Authors:** Aaron Lerner, Carina Benzvi, Aristo Vojdani

**Affiliations:** 1Chaim Sheba Medical Center, The Zabludowicz Center for Autoimmune Diseases, Ramat Gan 52621, Israel; carina.ben.zvi@gmail.com; 2Ariel Campus, Ariel University, Ariel 40700, Israel; 3Immunosciences Lab., Inc., Los Angeles, CA 90035, USA; drari@msn.com

**Keywords:** horizontal gene transfer, genetically engineered microorganisms, mobile genetic elements, regulation, autoimmune diseases, microbiome, dysbiome, gut, intestinal

## Abstract

Gut luminal dysbiosis and pathobiosis result in compositional and biodiversified alterations in the microbial and host co-metabolites. The primary mechanism of bacterial evolution is horizontal gene transfer (HGT), and the acquisition of new traits can be achieved through the exchange of mobile genetic elements (MGEs). Introducing genetically engineered microbes (GEMs) might break the harmonized balance in the intestinal compartment. The present objectives are: 1. To reveal the role played by the GEMs’ horizontal gene transfers in changing the landscape of the enteric microbiome eubiosis 2. To expand on the potential detrimental effects of those changes on the human genome and health. A search of articles published in PubMed/MEDLINE, EMBASE, and Scielo from 2000 to August 2023 using appropriate MeSH entry terms was performed. The GEMs’ horizontal gene exchanges might induce multiple human diseases. The new GEMs can change the long-term natural evolution of the enteric pro- or eukaryotic cell inhabitants. The worldwide regulatory authority’s safety control of GEMs is not enough to protect public health. Viability, biocontainment, and many other aspects are only partially controlled and harmful consequences for public health should be avoided. It is important to remember that prevention is the most cost-effective strategy and *primum non nocere* should be the focus.

## 1. Introduction

Many essential functions of the human body depend on the enteric symbiotic microbiota composition and biodiversity, essential components for human health. This intricate host–taxa relationship is a dynamic result of their long-term coevolution. This eubiosis harmonically maintains the host’s nutrition, metabolic passways, physiology, protective immune system and even behavior to the extent that we need them and cannot live without them. Greater phyla diversity is associated with microbiota resilience, sustained stability and greater ability to perform metabolic functions. The loss of microbiota phylogenic diversity and enhanced gut dysbiotic composition were associated with the Western lifestyle and several inflammatory, neurodegenerative, neurodevelopmental, infectious, metabolic, cancer and autoimmune diseases (ADs) that put human health at risk [1,2,3,4,5].

The primary mechanism of bacterial evolution is horizontal gene transfer (HGT), and new traits can be acquired through this mobile element exchange. Introducing GEMs might break the harmonized balance in the intestinal compartment [1,6]. The stable temperature, constant physicochemical conditions, continuous food supply, extremely high concentration of prokaryotic cells and phages, and plenty of opportunities for conjugation on the surfaces of host tissues and food particles represent one of the most favorable ecological niches for GEM-originated horizontal gene exchange of detrimental and harmful genetic sequences. Newly developed techniques of bacterial-mediated drug delivery have recently emerged using genetically engineered microbes aiming to locally deliver recombinant therapeutic proteins to the human gut. They are often called live biotherapeutic products, but they deliberately embed potential risks.

Entry of potentially unsafe GEMs into the human gut lumen can impact and change the selective pressures on gut microbiota and potentially contaminate the human microbiome with harmful genes exchanged horizontally. Presumably, the gut microbiome responds to these changes by genetic restructuring of gut populations, driven mainly via horizontal gene exchange. The objectives of the present narrative review are: 1. To reveal the role played by GEMs’ horizontal gene transfers in the changing landscape of the enteric microbiome eubiosis; 2. To expand on the potential detrimental effects of those changes on human health in general and autoimmune diseases in particular; 3. To warn against the impact of the new GEMs intestinal dwellers on many other pro- or eukaryotic cells, including changing the human genome. The following are some of the scientifically reported examples.

## 2. Numerous Harmful Mobile Genetic Elements (MGEs) Can Be Transferred to the Human Microbiome

Through their genomes, bacteria are subjected to rapid mutations and numerous rearrangements or HGT among and/or within bacterial species. Those MGEs, represented by bacteriophages, transposons, plasmids, and other pathogenic islands, represent a substantial amount of the microbial genome. Applying GEMs to the intestinal lumen can annulate the expression of beneficial genes while inducing the secretion of detrimental proteins. Alternatively, the GEMs can acquire the MGEs in the gut lumen. The following are various major and harmful clinical examples (Figure 1, Table 1).

**Antibiotic resistance genes** (ARGs) and **multidrug resistance (MDR) genes are the most reported** [1,7,8,9]. Less reported but not less important is the development of resistance of bacteria to phages [10,11], drug resistance to cancer therapy [12], resilience against antimicrobial defensive factors [13] and the MDR genes transfer along the food chain, including by contaminated and industrially processed nutrients [14]. The emergence of the resistome represents a worldwide health threat driven by the increasing unnecessary use of antibiotics and anticancer therapy. It occurs mainly by accumulating ARGs and MDR genes on MGEs, which is made possible by HGT [1,15]. Even the frequently consumed *Lactobacillus reuteri* was reported to carry ARGs [16,17]. The ARG does not originate only from human antibiotic consumption—antibiotic residue in food from animal sources can also drive the resistome [14,18]. Most recently, a high rate of ARG carried by Enterobacterales and diarrheagenic Escherichia coli in healthy donors screened for human fecal transplantation was noted [19]. The authors recommended multiplex PCR panels for stool donor screening. One wonders if the GEMs, said to benefit human health, are screened for ARGs or MDR genes.**Microbial-engineered enzymes** are an exponentially growing area that has become indispensable to processed food production, pharmaceuticals, and numerous other commercial goods [20]. Despite their beneficial effects on the processed food industries with increased production yields and “enhancing quality and sustainability” [21], multiple scientific publications are calling for a reassessment of their safety [22,23,24,25,26]. Intriguingly, a recent call was to reevaluate the GRAS definition allocated to various processed food additive ingredients. More reliable and updated approaches are offered to enzyme and other food nutritional categories for a more scientifically rigorous, sound and transparent application of the GRAS concept [27,28,29,30,31,32]. Moreover, a call to label, declare utilization and ensure consumer transparency regarding GEM enzymes is expressed in multiple scientific publications [28,33,34,35].Many nutritional components and nutrients are treated by GEM enzymes, resulting in post-translational modified proteins, turning naïve peptides into immunogenic complexes [2,30,32,36]. There are multiple examples of genetically engineered microbial enzymes; hence, one example will be expanded, namely the microbial transglutaminase (mTG).Microbial transglutaminase is a frequently used processed food additive, and its cross-linked complexes usage is expanding exponentially. The enzyme was classified as a processing aid and was granted the GRAS (generally recognized as safe) definition decades ago, thus avoiding a thorough assessment according to current criteria of toxicity and public health safety [24,25,26,37,38,39]. In contrast to the manufacturer’s declarations and claims, mTG and/or its transamidated complexes are proinflammatory, immunogenic, allergenic, pathogenic and potentially toxic, hence compromising public health [24,25,26]. Being a member of the transglutaminase family and functionally imitating the tissue transglutaminase to demidate or transamidate gliadin peptides, it was recently reported as a potential inducer of celiac disease [26,40,41,42]. In addition, its family member, the tissue transglutaminase, is a well-known inflammation inducer, fibrosis mediator and is heavily involved in sepsis [43,44]. Since mTG functionally imitates its endogenous member, one wonders if it contributes to those morbid conditions. Microbial transglutaminase and its docked complexes have numerous detrimental effects. Interestingly, in contrast to many publications showing the positive and beneficial aspects of mTG usage [45,46,47,48,49,50], there is evidence for the negative and harmful aspects of enzyme usage that might impact and compromise public health [25,26,28]. The debate between the GRAS category allocated by the FDA regulatory authorities for safe mTG consumption versus many critical scientific publications is ongoing. Several national regulatory committees have warned the public about the hazardous effects of mTGs [24,25,26]. In the case of mTG, it is possible for the gene responsible for its production to be transferred horizontally between microorganisms and even to eukaryotes [1,51,52]. Indeed, MGEs with mTG activity can potentially be transferred by HGT in between prokaryotes. Their presence in a gut luminal cellular compartment presents new opportunities for HGT, with the risk of inhabiting eukaryotic hosts [1,52,53]. One of the hypothetical scenarios is the acquisition of a classic microbial survival factor, such as a Trojan horse, against host self-defense barriers [1,25,26,54]. This gene exchange can happen through mechanisms like plasmid transfer or the incorporation of the transglutaminase gene into a MGE that can be transferred between bacteria. It is worth noting that the specific mechanisms and frequency of HGT for the mTG gene may depend on the particular microorganisms involved and the environmental conditions. The efforts to improve mTG production, thermostability and pH dependency by genetic engineering may do the opposite by enhancing the detrimental effects of the manipulated enzyme [55]. Finally, the fact that mTG is a bacterial survival factor can represent a significant positive selective pressure in the harsh, overcrowded luminal compartment [1,2], enhancing its HGT to other intestinal prokaryotic dwellers. It can be summarized that the mTG acts as a double-edged sword, protecting the microbes to survive in the gut lumen, hence compromising human health [24,25,26,54].The place of **probiotic** consumption should be highlighted in terms of their side effects. Drug resistance remains a universal threat, and the fad of probiotic consumption, many of which contain antibiotic-resistant elements, is a major and serious health concern [56,57,58,59]. In 2023, emerging issues in probiotic safety arose. Whole-genome sequencing should be implemented to detect virulence factors, toxins, ARGs and other detrimental MGEs [60]. The clear assignment of species and strain identity risks to vulnerable populations and the need for adverse event reporting are important topics to regulate.Engineered probiotics through gene editing is an emerging domain. Despite the reported clinical benefits for inflammatory bowel disease, infectious, tumor and metabolic diseases, tight regulatory measures are lacking [61]. Engineered and naïve probiotics compete with the luminal microbiome for nutrients or ecological niches and thus might affect the diversity and composition of intestinal microbiota. Human health can be more affected by their interaction with the luminal lipid metabolism [62]. Once again, consumer transparency, visible labeling and safety regulations are far from satisfactory.**Genetically modified (GM) plants** might possess beneficial traits like resistance to drought, pests and diseases, fighting climate change, improved agricultural and industrial production and enhanced nutrition. However, it also has a risky side to humans, animals and environmental health that should be regulated by national food security and regulatory authorities [63]. Mobile elements such as modified DNA can be laterally transferred to other recipients, spanning prokaryotes, eukaryotes and even to people [1,63]. More so, delaying tightened regulation risks facing increased GM plants, including genome-edited crops with deliberately altered and potentially harmful sequences [64,65,66]. A call for reconsideration before consumption [67], problematic and insufficient national legislation [68], risk of allergenicity [69] and consumer’s knowledge versus fears [70,71] are increasingly expressed concerning genetically modified food. Table 1 summarizes the harmful MGEs that potentially can compromise public health.

**Table 1 microorganisms-12-00238-t001:** MGEs harmful effect that can compromise public health.

MGEs	Potential Harmful Effects	References
Antibiotic resistance or multidrug resistance genes	Microbial antibiotic resistance Bacterial resistance to phages Drug resistance to cancer therapy Resilience against antimicrobial defensive factors Contaminated and industrially processed nutrients Potential entry to the human genome by HGT	[1,2,3,4,5,6,7,8,9,10,11,12,13,14]
Microbial-engineered enzymic genes with MTG as an example.	Post-translational modified proteins, turning naïve peptides into immunogenic complexes Complexes are proinflammatory, allergenic, pathogenic and potentially toxic, hence compromising public health Potential inducer of celiac disease	[2,11,24,25,26,28,33,34,36,41,42,72,73,74,75,76,77]
	MGE presence in a gut lumen presents new opportunities for HGT, with the risk of inhabiting eukaryotic hosts	[1,52,53]
	Transfer of microbial survival factors against host self-defense barriers	[1,25,26,54]
	Improved enzyme production, thermostability and pH dependency by genetic engineering might enhance the detrimental effects of the manipulated enzyme	[55]
Probiotics containing MGEs	Transfer of drug resistance Transfer of virulence factors, toxins, ARGs and other detrimental MGEs should be implemented	[56,57,58,59,60,61]
	Interference with the luminal lipid metabolism	[62]
Genetically modified plants	Modified DNA or other MGEs can be laterally transferred to other recipients, spanning prokaryotes, eukaryotes and even people.	[1,63]
	Genome-edited plants, like crops with deliberately altered and potentially harmful sequences, can invade the human microbiome or genome	[64,65,66,67,68,69,70,71]

GEMs—genetically engineered microorganisms, HGT—horizontal gene transfer, MGEs—mobile genetic elements, mTG—microbial transglutaminase, ARGs—antibiotic resistance genes.

## 3. GEMs’ Horizontal Gene Exchanges Might Induce Human Diseases

The engineered bacteria can produce modified proteins, peptides, nucleic acids, and other hazardous bioactive molecules that might drive various human pathologies and affect human health. Their products can potentially perturbate intracellular metabolic pathways, activate or turn off the expression of related genes and induce the synthesis of biologically active harmful molecules. In fact, current knowledge estimates that MGE represents more than one-half of the human genome [78].

The HGT sharing of DNA can unavoidably spread beneficial genes for prokaryotic survival, with mTG activity playing a role as genetic parasites across communities [79,80]. The resulting selective evolutionary pressure of the new dweller creates a large proportion of the variability acted on by luminal natural selection. This lateral gene exchange appears to be more ubiquitous in the human microbiome than previously described [1,80]. Consequently, unregulated GEMs might introduce new deleterious MGEs into the eubiotic gut lumen. 

The toxicity associated with GEM cells, which can limit their declared efficacy and enhance rapid clearance driven by the reactive immune responses stimulated by the bacterial load, might represent major drawbacks. Additionally, alteration of the composition, diversity, and disequilibrium in the gut eubiotic state might compromise human health. Autoimmune disease, inflammatory conditions such as diabetes, multiple sclerosis, rheumatoid arthritis, Inflammatory bowel diseases, obesity and even carcinogenesis might be promoted. 

The practical translation and implementation of GEMs are still hindered by potential harmful effects, as well as local legislation and regulations that limit clinical studies to use only bacteria without any genetic manipulations. Multiple challenges exist: 1. Limiting the spillage of genetically inserted genes over into the genomes of other microbes, prokaryotes or eukaryotic cells; 2. Ensuring the stability of the colonized engineered bacteria and the continuous production of the expected mobilome in the targeted tissues; 3. Efficient and helpful interactions with the enteric intestinal microbiome, intending to increase the microbiome–dysbiome ratio; 4. Locking the GEMs into the targeted tissues; 5. Clearing them once they accomplish their mission. Those challenges underscore the importance of ensuring the genetic stability of the foreign HGT cargo inside GEMs under laboratory and normal physiological conditions in vitro, ex. and in vivo [81]. Effective biocontainment measures are pivotal to preventing gene transfer in and out of the engineered microbes [82].

It is hoped that the health considerations of bacterial transgene HGTs will be thoroughly investigated and tightly regulated [65]. 

Following are some examples of the potential involvement of GEMs in chronic human diseases driven by disequilibrated gut homeostasis (Table 2):**Autoimmune diseases:** Various ADs are associated with specific [83,84,85,86] or pathobiont [87]. Type 1 diabetes, multiple sclerosis, celiac disease and psoriasis are some of them [88]. The above-cited mTG is also associated with AD evolution [24,25,26,34,36,54,73,77]. Intriguingly, the cross-reactive antibodies and sequence similarity between microbial transglutaminase and human tissue antigens were recently reported [54]. Six human epitopes were connected to ten different ADs. The newly described molecular mimicry pathways further strengthen the mTG-ADs pathologic interplay.**Neurodegenerative conditions:** Understanding the involvement of gut dysbiosis and pathobiosis is in its infancy; however, increased knowledge is starting to appear, thus strengthening the gut–brain axis [28,89]. By perturbating enteric eubiosis and/or its beneficial secreted metabolome, the GEMs can potentially drive neuro-inflammatory/degenerative diseases [28,73,89,90,91]. Interestingly, those GEMs included transposable elements that might drive neurodevelopmental and neurodegenerative Disorders [92].**Metabolic diseases:** All the components of the metabolic syndrome are related to a perturbated gut microbiome, hazardous mobilome, and disbalance of a fine synergistic luminal homeostasis [93,94,95,96]. Harmful proteinomes and metabolomes, increased intestinal permeability, post-translational modification of naïve peptides to immunogenic ones, cross-reactive autoantibodies, sequence similarity, molecular mimicry, bacterial fragments blood translocation and some other auto-immunogenic pathways might drive GEM involvement in metabolic conditions [2,85,96,97].**Allergic conditions:** Food allergy is highly related to intestinal dysbiosis, and eubiotic equilibrium might protect allergy patients [98]. Natural or GEM probiotics, prebiotics, synbiotics and potentially fecal microbiota transfer are increasingly being investigated to alleviate allergic reactions. Those trials should be controlled and regulated; they impose a variety of challenges, aiming to improve the reliability and predictability of the allergenicity risk assessment. A clear safety objective that addresses new GM biotechnologies is greatly needed as safety assessments to ensure that allergenic risks of foods are avoided [99].**Cancer induction or therapy**: HGT occurs between prokaryotes and eukaryotes [100] and microbes, viruses or fungi are related to human cancer induction [101]. One recent example is the engineered *E. coli Nissle* 1917 involvement in colorectal cancer [102]. In contrast, prokaryotes are increasingly reported as key actors in cancer immunotherapy, applying engineered biotechnologies to combat spreading by metastases [103,104]. The potential HGT of carcinogenic constituents, from unicellular prokaryotes to multicellular tissues, including human cancer cells, requires urgent tightened control and regulatory measures on GEMs [1,105,106,107]. Recently reported examples of bacterial DNA were confirmed in lung, pancreatic, breast, bone and colorectal cancers and malignant melanomas [107]. Several mechanisms of microbial DNA integration into the human genome and cancer induction were suggested. One of those is to increase proto-oncogenes or suppress tumor suppressor gene expression in the human genome [107]. However, this can be a self-perpetuating vicious cycle, as recently noted by Yangyanqiu and Shuwen: “The damage caused by bacteria to human DNA, such as inducing DNA breaks, regulating gene expression by epigenetic modifications, and causing genome instability, can facilitate the integration of bacterial DNA into the human genome” [107]. In addition, microbial enzymes, like recombinases, can facilitate the site-specific insertion of MGEs into bacterial genomes, thus loading the intestinal microbiome and risking human cells for large-payload genome insertion [108]. Even prebiotic oligosaccharides intake might aggravate DNA damage induced by colibactin-producing gut microbes [109,110]. Interestingly, a high-fiber diet and indigestible prebiotic saccharide are offered to prevent colorectal cancer. In contrast, the authors suggested that the enhanced progression of colorectal cancer operating through cellular senescence, double-strand break induction in cultured cells, and chromosomal abnormalities depends on prebiotic oligosaccharides. Future studies are necessary to resolve this discrepancy.Nevertheless, the topic of microbial genes integrated into the human genome is an ongoing hot topic. Its contribution to the evolution of eukaryotic genomes remains high [1,107,111]. Since prevention is the most cost-effective way to fight cancer or other human chronic diseases, tightly regulating and controlling GEMs and avoiding the entry of MGEs into the microbiome or human genome represent the most rewarding means to protect people from those morbid and mortal conditions.**Neurodevelopment and behavior**: Explicit emotion regulation and cognitive control govern executive functions and mental health throughout the entire lifespan. The intestinal microbiota represents a potential biomarker for the risk of mental and behavioral morbidities. Basically, gut eubiotic diversity and synergistic composition affect brain function, thus playing a pivotal role in emotional processing [28,73,89,112,113,114,115,116]. Recently, the following neuropsychiatric conditions were reported to be dysbiotic-dependent: Alzheimer’s disease, attention deficit hyperactivity disorder, amyotrophic lateral sclerosis, anorexia nervosa, bipolar disorder, generalized anxiety disorder, major depressive disorder, multiple sclerosis and schizophrenia [117]. The microbiome–gut–brain axis plays an essential role in regulating neurodevelopment, brain metabolism and behavior. Tryptophan, the precursor to serotonin, short-chain fatty acids, GABA, acetylcholine, histamine, bile acids, 5-amino valeric acid, taurine and spermine are some of the microbiome-originated neurotransmitters and metabolome that affect brain physiology, human behavior or pathology [89,113,118,119]. Introducing less-regulated GEMs or their foreign mobilome to the luminal compartment might disrupt the evolutionary equilibrium of the enteric inhabitants.**Female and male infertility:** Most recently, genetically proxied intestinal microbes were found to have potential causal effects on females and males [120,121]. This additional potential risk might affect future generations of geo-epidemiology and many other public aspects of life worldwide. One could wonder about the potential impact of deleterious MGE entry into the equilibrated intestinal microbiome on the above-cited chronic human diseases. Table 2 summarizes the potential involvement of GEMs in chronic human diseases driven by perturbated gut homeostasis.
microorganisms-12-00238-t002_Table 2Table 2The Role of GEMs in Chronic Diseases Linked to Disrupted Gut Homeostasis.Chronic Disease CategoryDisease ExamplesReferencesAutoimmune diseasesType 1 diabetes, MM, celiac disease, MG, GBS and psoriasis [24,25,26,77,83,84,85,86,87,88]Neurodegenerative conditionsAlzheimer’s, Parkinson’s, autism, schizophrenia, ALS and MM [28,73,89,90]Metabolic diseasesType 1 diabetes, cardiovascular, hyperlipidemia, obesity and liver steatosis [1,2,11,28,34,36,75,79,80,85,93,94,95,96,97,122]AllergyFood allergies [98,99]Cancer lung, pancreatic, breast, bone and colorectal cancers and malignant melanoma [1,100,101,102,103,104,105,106,107,108,109,110,111]Neurodevelopment and behaviorBipolar, depression, anxiety, ADHD, migraines and headaches  [28,73,89,112,113,114,115,116,117]infertilityFemale and male infertility [120,121]Abbreviations: MM—multiple sclerosis, MG—myasthenia gravis, GBS—Guillain–Barré syndrome, ALS—amyotrophic lateral sclerosis, ADHD—attention deficit hyperactivity disorder.

## 4. The New GEMs Can Change the Long-Term Natural Evolution of the Enteric Pro- or Eukaryotic Cells Inhabitants

Genome editing is an indispensable tool for modulating specific functions of individual genes or changing the expression of important genes in the cells and whole organisms. CRISPR (clustered regularly interspaced short palindromic repeats)-Cas (CRISPR-associated) is a pivotal prokaryotic adaptive immune machinery that protects the microbiome from invading viruses and plasmids [123,124]. The CRISPR/Cas system represents a major driving force and a game-changer in the life science revolution in the 21st century due to its advantages in genome editing and regulation. On the contrary, despite the advantages and the embedded challenges, using this technology for gene editing in GEMs might bring risks and devastating consequences on the enteric prokaryotic inhabitants and the entire human genome [125]. Genetically engineered probiotics or synthetic microbial consortia should be regulated as drugs, not as less-controlled food supplements [126]. Their gut delivery could bring unexpected consequences, causing biosafety problems that should be addressed and overcome by regulatory authorities. The same applies to genetically modified food after entering the human body [127]. Risks to fetuses, plant toxin production, HGT to the enteric microbiome, spreading ARG and MDR delivery and allergenic reaction induction are some of the reported risks [67,69,127]. Intriguingly, a comparable risk exists in the usage of engineered fungi [128]. There is no doubt that improved and validated ways of safely using genetically modified nutrients should be implemented. The labeling of genetically modified ingredients to satisfy public transparency should be adopted. In summary, tightening the regulation of GEMs and engineered plants is urgently needed. Table 1 and Figure 2 summarize the potential involvement of GEMs in chronic human diseases driven by perturbated gut homeostasis.

## 5. The Worldwide Regulatory Authority’s Safety Control of GEMs Is Not Enough to Protect Public Health

Maximizing the safety of the GEMs is important, necessary and indispensable. Scientific critics are increasingly raising concerns regarding various safety issues in GEM development, clinical applications and usage. The following is a summary of those concerns and warnings.

**Biocontainment** should be controlled and regulated for real-world applications [129,130,131,132]. This can be achieved by biocontainment genetic circuits, auxotrophic mechanisms and reliance on synthetic amino acids or protein designs. These means will help to prevent the spread and persistence of GEMs in the environment. Sensors for tight biocontainment will ensure viability control.**GEM genetic instability** to enhance their stability in the gut compartment. This will reduce the probability of loss or gain-of-function mutations [129,130]. Their limited or lack of luminal colonization capacity and easy eradication by routine antibiotic intake administration might limit GEM efficacy.**Individual inherent microbiome** variations may dilute the GEMs’ intestinal functionality [130]. Thus, predicting the long-term engraftment of modified bacteria within any given patient endogenous population might be difficult to achieve, resulting in a kind of personal medicine.**Competition with stable and long-term eubiotic communities** might adversely perturbate the delicate and fragile balance of the gut ecosystem [131].**Uncontrolled growth of the GEMs** within the human gut [130]. Biocontainment strategies are essential prior to the clinical application in order to avoid gut dysfunction, intestinal inflammation [132] or pathogenic infection [81].**GEM-induced metabolic abnormalities and their toxic effects** should be fully evaluated before their in vivo clinical usage [132].**Controlling and limiting the viability** of the inhabitant GEMs is necessary. Live biotherapeutic-engineered microbes can induce unwanted detrimental dysfunction and break gut homeostasis, resulting in microbiome disruption and potential organ pathogenicity [123]. Genetic “kill-switch” strategies designed to lyse the cell when triggered are crucial. Alternatively, GEM pathogenicity should be mitigated by gene knockout or mutated virulence genes [123,132].**Clearing the foreign-modified bacteria** after accomplishing its therapeutic effects is a key task. Alternative selection markers, biocontainment and homologous DNA usage were applied to avoid potential environmental transmission and purge the residual foreign bacteria [132,133].**Controlling the microbial production pathways.** Dynamic regulation is a strategy to control the production of key molecules. Transcription factor-based biosensors for the dynamic regulation of the final product were recently reported [134,135]. By detecting and following the presence of the synthesized molecule and triggering the inhibition or activation of targeted genes in the metabolic pathway, the biosensors might help to tighten the regulation.

In summary, clinical microbial live biotherapy has major safety hurdles to overcome before it can be routinely used, and appropriate regulatory enforcement must be made available. Current safety practices related to GEM evolution, clinical applications and use are far from satisfying public health demands. Several emerging and sophisticated techniques might help track those harmful GEMs, understand the enigma of potential DNA relocation and develop more comprehensive regulatory strategies regarding health benefits. 

Indeed, chromosome conformation capture and methylome analyses [136], the bioinformatic pipeline (Xenoseq) application [137], the use of organoids and the microfluidic ‘Gut-on-a-chip’ technique were suggested [138]. Most recently, novel computational strategies were reported to merge theoretical models with experimental methods [139]. Combining those approaches enables numerous strains and GME transfer to be studied, both in vivo and in vitro, thus mimicking the intricacies of luminal-associated dysbiotic and pathobiotic–human morbidity relationships. In 2012, the U.S. Environmental Protection Agency issued a summary on the Regulation of Genetically Engineered Microorganisms Under FIFRA, FFDCA and TSCA [140], which did not address the burning issues of the potentially harmful effects of the HGT of GEMs in the human intestinal compartment. These issues must be addressed. The risk of mutation and the transfer of genetic material are cause for concern and calculated caution [141,142].

## 6. Conclusions

Along with in-depth analyses, the present information unveils the importance of regulating the GEMs pre-inhabiting our enteric harmonized eubiome. Newly introduced genetic cargo can potentially perturbate the symbiotic and fine, hence, fragile, enteric synergistic homeostasis. The foreign, non-self MGEs represent a potential threat to human physical and mental health.

However, further comprehensive, well-designed and evidence-based studies are required to draw more solid conclusions regarding the tight regulation of GEMs, their mechanisms of action and contemporary and evolutionary potential detrimental impacts that aim to prevent their harmful effects on human beings. Widespread use of natural and genetically engineered intestinal biotics should be halted, and public labeling and clear transparency should be instituted. Regulatory guidelines for gut GEM usage must be backed up by basic and clinical research. A more holistic comprehension of gut HGT-dependent eubiotic–dysbiotic balance, along with multiple environmental and lifestyle factors, is necessary to better manage and prevent the drawbacks of widespread GEM usage. This narrative review encourages regulatory authorities worldwide to take a more holistic and aligned approach to the risk evaluation and regulatory oversight of GEM-produced food ingredients, immune factors, enzymes and any category of food substances that can enable safe and sustainable consumer food choices and consumption.

Despite their proven therapeutic benefits, synthetic microbial biotherapeutics have several safety hurdles to overcome to reach widespread usage and consumer acceptance. Extensive studies are required to explore the multi-directional communication between gut homeostasis and the newly introduced GEMs, which might help researchers understand the newly engineered inhabitant effects on public physical health and mental behavior. It is important to remember that prevention is the most cost-effective strategy and *primum non nocere* should be the focus.

## Figures and Tables

**Figure 1 microorganisms-12-00238-f001:**
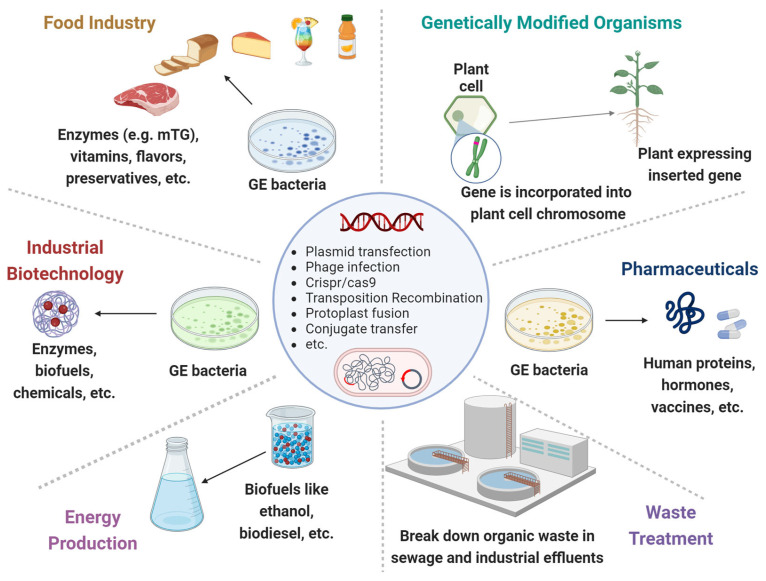
**Genetically engineered microorganisms (GEMs) applications.** GEMs have a wide range of applications across various fields due to their versatility and the precision of genetic engineering techniques. **Food Industry**: Production of vitamins, flavors, enzymes, and preservatives. They can help in improving the nutritional value, taste, and shelf-life of food products. **Agriculture**: Promote plant growth, increase nutrient uptake, and protect plants from pests and diseases. **Medicine and Health Care:** Cost-effective production of pharmaceuticals, including insulin, growth hormones and vaccines. **Waste Treatment:** Break down hazardous substances like oil spills, heavy metals and other toxic chemicals. **Energy Production:** The production of biofuels like ethanol and biodiesel. **Industrial Biotechnology**: Improve chemical production to increase yields and reduce environmental impacts.

**Figure 2 microorganisms-12-00238-f002:**
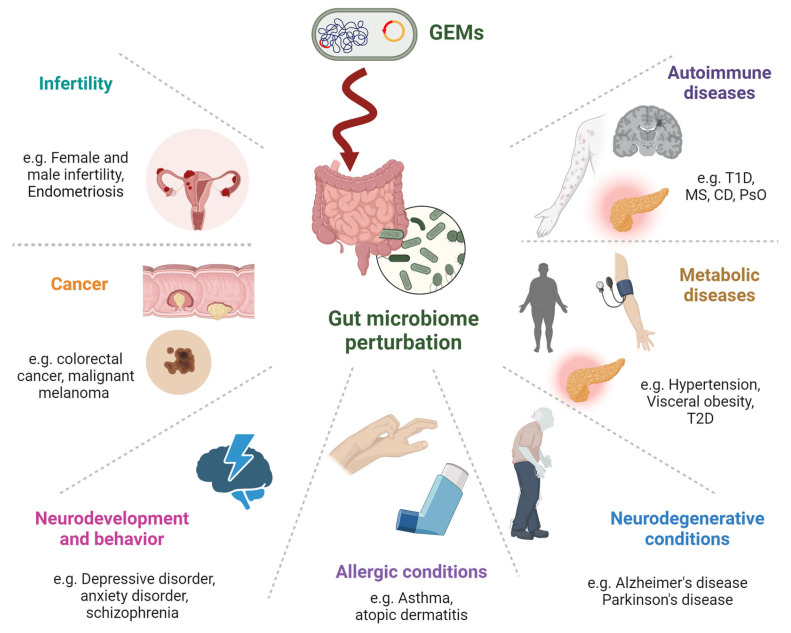
**Potential involvement of GEMs in chronic human diseases related to disrupted gut homeostasis covers a broad spectrum of conditions. Autoimmune Diseases:** affecting immune tolerance and increasing inflammation. **Metabolic Diseases:** Influence the metabolism of lipids, carbohydrates, and other substances, which can lead to obesity, type 2 diabetes, and metabolic syndrome. **Neurodegenerative Conditions:** Through the gut–brain axis, gut bacteria perturbation can produce neurotoxic substances, affecting neurodegenerative diseases like Alzheimer’s and Parkinson’s. **Allergic Conditions:** This might influence allergic diseases by modifying the gut microbiome, which is known to play a role in immune responses. **Neurodevelopment and Behavior:** Via the gut–brain axis, gut microbiome compounds can affect mood, cognition, and behavior. **Cancer:** Microbial genes that are integrated into the human genome can potentially induce carcinogenic constituents. **Infertility:** Emerging evidence linking gut health to reproductive health. This might involve the modulation of hormone levels, inflammation, and overall metabolic health.

## Data Availability

Not applicable.

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
