# Peer review of "The Potential Harmful Effects of Genetically Engineered Microorganisms (GEMs) on the Intestinal Microbiome and Public Health"

_microorganisms, 2024, doi:10.3390/microorganisms12020238_

Round 1
Reviewer 1 Report
Comments and Suggestions for Authors
The article addresses some key facets of the latest research on the potentially harmful effect of GEMs on intestinal microbes and public health. It covers relevant content; however, certain queries need to be addressed through a comprehensive article revision.
1. The basic concern is that the article is monotonous; enhance the quality of the article by using some tables and figures.
2. For the single statement, authors used multiple references, for example, statements in lines 111, 114, 123, 233 etc. The author should support the assertion with 2 or 3 relevant references, and the rest of the information can be provided as a table.
3. Focus on the conclusion and future perspective, it might seem unconventional to include references in the conclusion; however, refrain from doing so. Craft the conclusion and perspective based on the study's finding
Comments on the Quality of English LanguageMinor english editing
Author Response
Response to reviewer 1
Thanks for your valuable comments. All of them were taken care and are highlighting in yellow in the text.
- The basic concern is that the article is monotonous; enhance the quality of the article by using some tables and figures. Table 1 was added. The present review contains 2 figures and 2 tables
- For the single statement, authors used multiple references, for example, statements in lines 111, 114, 123, 233 etc. The author should support the assertion with 2 or 3 relevant references, and the rest of the information can be provided as a table. Done. Please see table 1 and 2
- Focus on the conclusion and future perspective, it might seem unconventional to include references in the conclusion; however, refrain from doing so. Craft the conclusion and perspective based on the study's finding. Done. References were deleted and conclusions were more focused.
We appreciate the reviewer comments and suggestions. The present version is upgraded following the valuable review.
Reviewer 2 Report
Comments and Suggestions for Authors
This review is dedicated to Genetically Engineered Microorganisms on the intestinal microbiome and public health. In my opinion, the review is well written and fairly fully presents the problem to which it is devoted. In my opinion, the review can be published without changes.
I believe that this review is interesting because it examines recent work on this topic. This will be useful to researchers working in this area. This is the first review summarizing recent work on this topic. The authors used a PubMed search, which is a common methodology when writing reviews. The review contains only one drawing, but it is made of very high quality. I don’t really understand why it’s included in the supplementary. In my opinion, it could be in the main text. Otherwise, the review may be published as presented.
Author Response
The response to reviewer 2 is attached

Round 2
Reviewer 1 Report
Comments and Suggestions for Authors
The article has been modified substantially based on the raised query. I recommenr to publish the atricle.
Comments on the Quality of English LanguageNA